# Optimization of the Outlet Flow Ratio of Mini-Hydrocyclone Separators Using the Full Factorial Design Method to Determine the Separation Efficiency

Hyung-Wook Yi [1,2] , Je-Young Kwon [1,2] , Yu-Wool Lee [2] and Myung-Chang Kang [2,*]

1 Hong Sung Co., Ltd., Yangsan-si 50576, Korea; ramjee@nate.com (H.-W.Y.); hungrybird@naver.com (J.-Y.K.)
2 Graduate School of Convergence Science, Pusan National University, Busan 46241, Korea; ducl3612@naver.com
* Correspondence: kangmc@pusan.ac.kr; Tel.: +82-51-510-7395

**Abstract:** Cyclone separators are widely used to eliminate particles flowing through pipelines in equipment from various industrial processes. Unlike general filters, cyclone separators can constantly and effectively eliminate foreign substances present in the fluid flowing through the equipment. In this study, we fabricated mini-hydrocyclone separators using the 3D printing method for application in the steam and water analysis system (SWAS) in a thermal power plant instead of the conventional strainer filter. The gravimetric method was used to measure the separation efficiency of the hydrocyclone separators and compare the weights of the sludge discharged from the underflow and overflow outlets. The outlet flow ratio was optimized by adjusting the diameters of the spigot and vortex finder of the separators, which influenced the outlet flow rate. To apply the gravimetric method more objectively, the optimum values of the diameters of the vortex finder and spigot with an outlet flow ratio of 1 were determined using full factorial design (FFD) in the design of experiments (DOE). The obtained values were verified through numerical analysis using the ANSYS Fluent software. Furthermore, after fabrication of the mini-hydrocyclone separators using an SLA-type 3D printer, we conducted a numerical analysis, and the results were compared with that of the actual experiment. It was observed that the use of FFD can effectively optimize the desired outlet flow ratio in the mini-hydrocyclone separator. In addition, the changes in the outlet flow ratio do not affect the separation efficiency of the cyclone separators.

**Keywords:** hydrocyclone; cyclone separator; full factorial design; 3D printing; outlet flow ratio

## 1. Introduction

Cyclone separators are mechanical devices that separate foreign substances from fluids using centrifugal force. They are widely used in household cleaners and air purifiers owing to their simple structure and can be manufactured easily, as they do not require separate driving devices [1]. Cyclone separators can be classified as hydrocyclone and gas cyclone separators. Gas cyclone separators are used to separate solids and liquids from gases [2]. Hydrocyclone separators are used to separate liquids and solids. First, water comprising solid particles is injected through the inlet of the device, and the particles and water separate owing to the centrifugal force. The large and heavy particles are eliminated from the underflow outlet, whereas the small and light particles are eliminated from the overflow outlet [3].

Considering that the cooling tower and steam boiler in thermal power plants require water for operation, the water quality is analyzed in real time using the steam and water analysis system (SWAS) to protect the equipment. Generally, the water samples received by SWAS from different equipment through pipelines are contaminated with foreign substances. Therefore, in this study, we develop a mini-hydrocyclone to replace the strainer-type filter used in the SWAS equipment in nuclear and thermal power plants to

eliminate foreign substances flowing through the pipelines and prevent failure of the water quality analyzer to improve the overall analysis efficiency. Figure 1 is a schematic diagram of the SWAS of a thermal power plant. First, the process water is sent to the first chiller for cooling. Then, the water is supplied to the hydrocyclone for sludge removal. Finally, the water passes through a second chiller and is sent to the water quality analyzer.

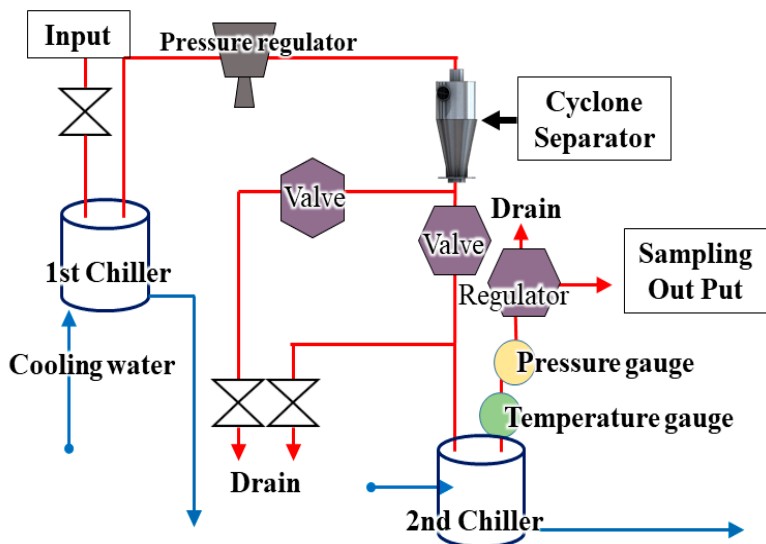

**Figure 1.** SWAS schematic diagram with cyclone separator applied.

In this study, we designed the mini-hydrocyclone separators based on the high-efficiency Stairmand model while focusing on changing other factors and retaining the diameter of the cylindrical body depending on the suggested flow rate. Figure 2 is a concept diagram of hydrocyclone. Several published studies have determined the performance of a cyclone separator from the relative proportions of its shape, based on indicators such as the inlet dimension, inlet pressure, spigot diameter, vortex diameter, cylinder diameter, and cone length [2,3].

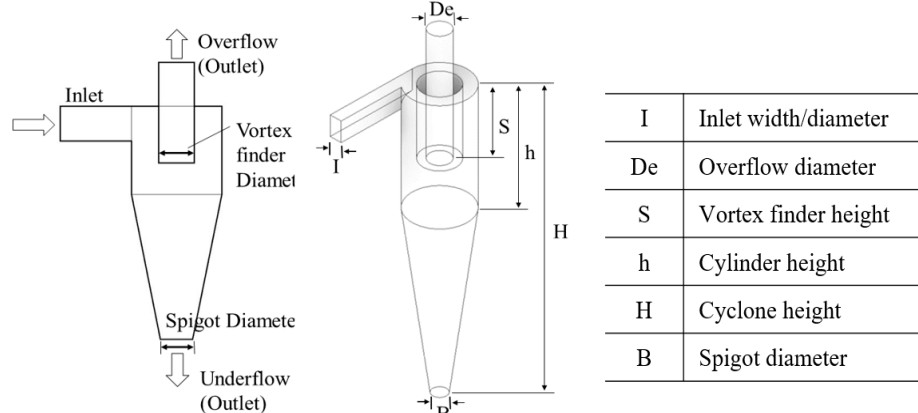

**Figure 2.** Hydrocyclone design concept and parameters.

Several studies have reported that the separation performance and characteristics of cyclone separators are highly dependent on the geometric shape of the separator. In particular, the diameters of the vortex finder and spigot most significantly affect the outlet flow rate [4–6]. The following examples are research cases by experiment. Chu made various changes to the configuration of the hydrocyclone [7]. Xiang studied the performance change of hydrocyclone owing to the change in the diameter of the underflow

outlet [8]. Chuah conducted research on the change in the diameter of the underflow outlet of the hydrocyclone using CFD [9].

However, performance indicators that can be physically adjusted are extremely limited for mini-hydrocyclone separators owing to their small size. Manufacturing mini-hydrocyclone separators using conventional methods is highly complicated and difficult, as they are fabricated using the 3D printing method [10]. Notably, 3D-printed mini-hydrocyclones have been recently optimized to separate fine particles with low flow rates [11,12].

The shape of the cyclone separator should be changed to optimize the outlet flow ratio based on demands. However, these changes should be limited because they can significantly affect the efficiency and performance of the cyclone separator and deviate from the original purpose. In this study, we attempted to determine the targeted outlet flow ratio by adjusting the diameters of the vortex finder and spigot while maintaining other geometrical parameters.

Previous research on the hydrocyclone was conducted by evaluating the outlet flow ratio of the hydrocyclone through experiments or numerical analysis. However, we used the FFD method to solve the problem.

The full factorial design (FFD) method is a rationale for the design of all the experiments and is most frequently used for optimization and factor selection [13–16]. The experiment was performed by altering other experimental variables while fixing one variable. However, it is difficult to determine the correlation between these factors considering the time required to optimize the experimental conditions [17]. FFD creates all the possible combinations of levels that each design variable can exhibit and analyzes the influence of interactions between major factors, making it suitable for identifying the characteristics of factors and obtaining an optimal combination [18].

Therefore, in this study, the variable values for the diameters of vortex finder and spigot were optimized using the FFD method to determine the value with an outlet flow ratio of 1. The results were verified through numerical analysis using the ANSYS Fluent software (Version 16.1, ANSYS Inc., Canonsburg, PA, USA). Finally, we fabricated the mini-hydrocyclones with an optimized outlet flow ratio using 3D printing and compared results with that of the previous experiment.

## 2. Analysis and Experiment

### 2.1. Mini-Hydrocyclone Fabrication and Experiment

In this study, we designed the mini-hydrocyclone separators based on the high-efficiency Stairmand model (Stairmand, 1951). We used the project SLA 6000 HD (3D SYSTEMS, Santa Clarita, CA, USA) 3D printer and polypropylene-like (PC-Like) material. As shown in Figure 3, PC-like materials are transparent, which helps directly observe the flow inside the cyclones from the outside and have sufficient strength to be used for fabricating cyclone separators. The precision of the fabricated mini-hydrocyclone separator was 0.025–0.05 mm per 25.4 mm.

Figure 4 shows the configuration of the experimental equipment. $AL_2O_3$ (50 g) with a particle size of 14.5 μm was mixed in 2000 mL of water using an agitator. We used $AL_2O_3$ particles as a substitute for sludge, an impurity found in pipelines. This is because $AL_2O_3$ can sufficiently represent the specific gravity values of various foreign substances existing in the pipelines considered in this study. The mixture was then supplied to the inlet at a flow rate of 2000 mL/min using a BT600-2J medium-high flow rate peristaltic pump (Longer, Baoding, China). The fluids from the underflow and overflow outlets were collected in separate beakers. Once the collected mixtures were dried using a dryer, the obtained $AL_2O_3$ particles were weighed and compared using the gravimetric method.

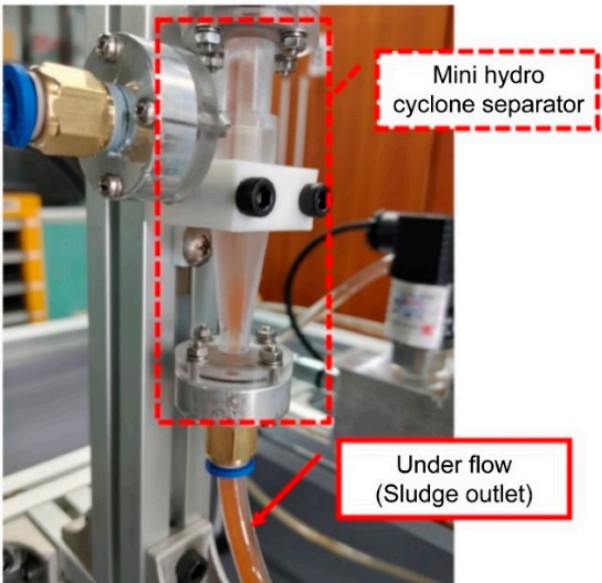

**Figure 3.** Owing to the transparency of the PC-Like material, the flow inside the cyclone separator could be observed from the outside.

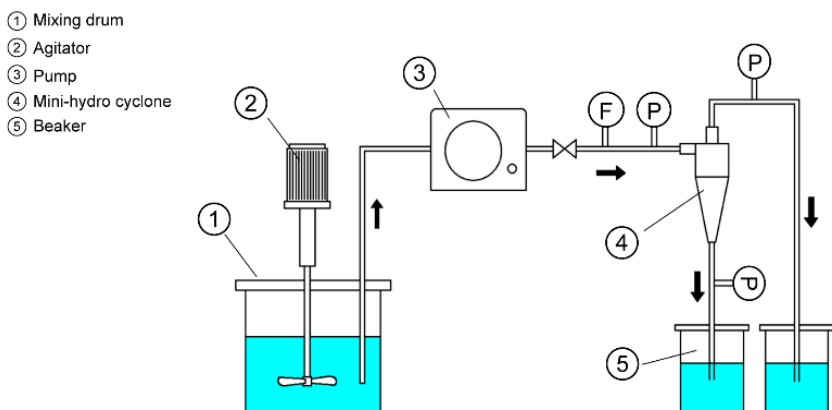

**Figure 4.** Schematic representation of the separation efficiency measurement experimental device using the fabricated mini-hydrocyclone separator.

As shown in Figure 5, we originally designed and fabricated three types of mini-hydrocyclones. Table 1 summarizes the results of model A, which showed the best separation efficiency. The gravimetric method was calculated using the following equation:

$$E_{sp} = \frac{W_{under}}{W_{under} + W_{over}} \qquad (1)$$

where $E_{sp}$ is the separation efficiency, and $W_{under}$ and $W_{over}$ are the underflow and overflow sludge weights, respectively. Although 50 g $AL_2O_3$ was added, a particle loss of approximately 2 g on average was observed, which was excluded from calculations. At this time, the outlet frow ratio was 0.54, which indicated a high underflow rate. Therefore, there was a severe imbalance between overflow and underflow.

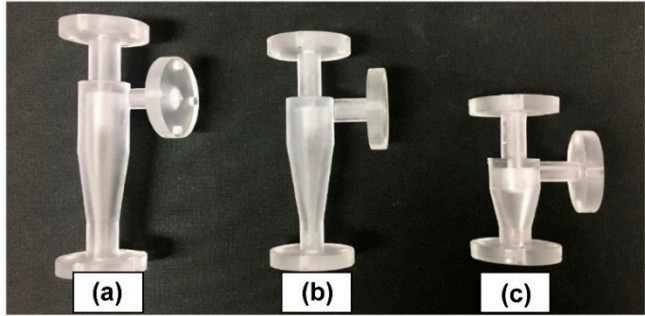

**Figure 5.** The three types of mini-hydrocyclones fabricated using 3D printing: (**a**) high-efficiency Stairmand type #1, (**b**) high-efficiency Stairmand type #2, and (**c**) standard Stairmand type.

**Table 1.** Separation efficiency results of model A, measured using the gravimetric method.

| Total Overflow Volume | Total Underflow Volume | Overflow Sludge Weight | Underflow Sludge Weight | Separation Efficiency |
|---|---|---|---|---|
| 700 mL | 1300 mL | 3.05 g | 45.10 g | 93.67% |

Results revealed that the fabricated mini-hydrocyclone separators exhibited satisfactory separation efficiency of over 90%. However, the majority of water flow discharge came from the underflow outlet, where it was discarded.

Therefore, the following two limitations were observed. First, because the flow in the SWAS equipment should be an overflow, excessive flow to the underflow outlet can cause a problem when using the equipment. Second, because the efficiency was measured using the gravimetric method in this study, an optical illusion exists that separation is better considering that the underflow flow rate is significantly higher than that of the overflow. Therefore, a cyclone separator with the same outlet flow rate would be required for a more accurate evaluation. In this study, the concept of outlet flow ratio was introduced for the efficiency evaluation of the hydrocyclone using the gravimetric method. The outlet flow ratio, which is dimensionless, is calculated depending on the weight of the particles separated from the underflow and overflow outlets, given as

$$C_{of} = \frac{V_{of}}{V_{uf}} \tag{2}$$

where $C_{of}$ is the outlet flow ratio, and $V_{of}$ and $V_{uf}$ are the overflow and underflow volumes, respectively.

### 2.2. Full Factorial Design (FFD)

Using the gravimetric method, we calculated the separation efficiency from the weights of the sludge from the two outlets. If the difference in the flow rates of the two outlets is too large, it would be difficult to apply the gravimetric method. Therefore, the flow rates of the overflow and underflow outlets should be the same, that is, the outlet flow ratio should be close to 1.

An outlet flow ratio of 1 indicates that the flow rates of both outlets are the same. When the outlet flow ratio is greater than 1, the overflow rate is higher. Conversely, when the outlet flow ratio was smaller than 1, the underflow rate was higher. Table 2 summarizes these relationships. To determine the diameters of the vortex finder and spigot to satisfy the condition of outlet flow ratio = 1, FFD was used among the DOEs. The experiment was repeated three times at level 2 using two variables; the details are listed in Table 3. The analysis was performed using Minitab19 statistical software, and a total of 12 runs were generated. The maximum and minimum values were selected while determining the range of variables to specify the range of values. As the physical size of the mini-hydrocyclone was small, the range of the maximum and minimum values was approximately 0.5 mm.

**Table 2.** Outlet flow ratio value.

| Outlet Flow Ratio | Overflow | Underflow |
|---|---|---|
| Value > 1 | Big | Small |
| Value = 1 | = | = |
| Value < 1 | Small | Big |

**Table 3.** Level values for each factor applied to FFD.

| Factor | Unit | Low Level | High Level |
|---|---|---|---|
| Vortex finder diameter | mm | 3 | 3.5 |
| Spigot diameter | mm | 2.5 | 3 |

The coefficient of determination (R-squared), which is a measure of how close the observations are to the regression line and measures the fit of the regression line, must be 70% or higher to determine the optimal conditions.

Equation (3) shows the regression equation using the above-estimated overflow rate coefficient.

$$OF = 76.3333 + 7.33333 \times VF - 24.0000 \times SP + 1.33333 \times VF \times SP \tag{3}$$

where $VF$ is the vortex finder diameter, and $SP$ is the spigot diameter.

Figure 6 shows the Pareto chart of standardized effects. While the (A) vortex finder and (B) spigot diameter were significant in a Pareto chart, which visualizes the relative magnitude of the effects by variables, the interaction between them was not significant.

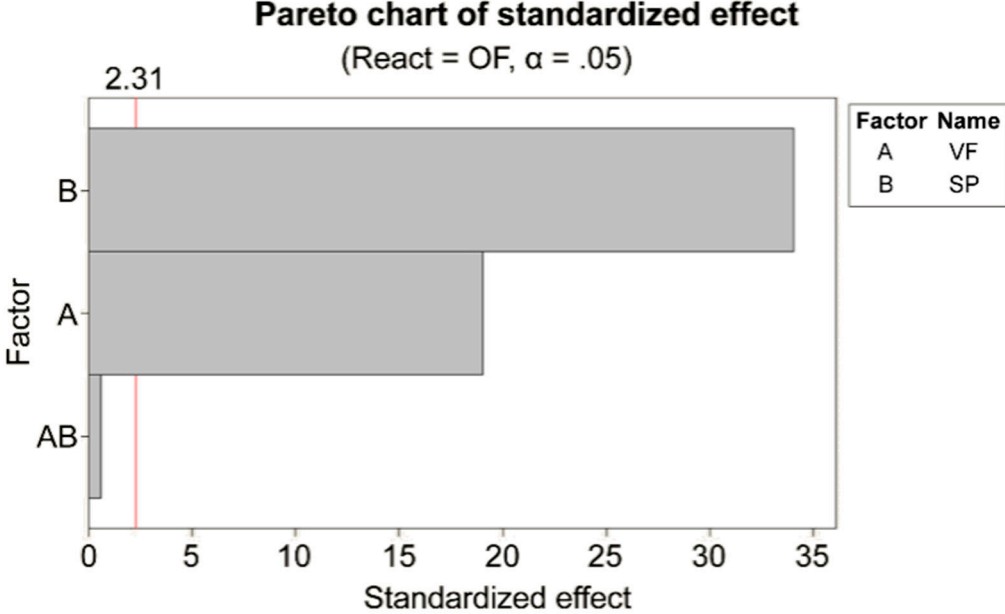

**Figure 6.** Pareto chart of the standardized effects on the overflow of (A) vortex finder and (B) spigot diameter confirmed by Pareto chart. (AB) is the interaction between vortex finder and spigot.

### 2.3. Numerical Methods

A numerical analysis was performed using the ANSYS Fluent software to verify the acquired results and match the outlet flow ratio of 1 in the FFD. The flow inside the hydrocyclone was mixed with laminar and turbulent flows. For the numerical analysis of complex flow, transient state and the Reynolds stress model (RSM) for turbulence were used. Among many available turbulence models, RSM is an appropriate model for the interpretation of flow with a strong swirl.

Air core formation has been investigated in many hydrocyclones. Hydrocyclones form a central air core that extends over the entire length of the hydrocyclone. Air is sucked into the core at the underflow discharge. The formation of the air core in any hydrocyclone is considered to be an indication of vortex stability [19]. The internal flow of the hydrocyclone represented a condition where water, particles, and air coexisted. Therefore, VOF was selected among multiphase flow interpreting models. Additionally, both the water and particles were considered as a continuum flow. Tracking is realized by the writing force equilibrium equation of a solid particle assuming different types of force according to the complexity of the model. This approach is only suitable when a reasonable number of particles are tracked and the suspension is diluted (max 5–10% mass fraction of the solid phase).

Table 4 summarizes the analysis conditions. Water was used as the fluid, and the physical properties provided by the Fluent software were used as the constants. The flow inside the hydrocyclone separator was assumed to be in a steady state, and the temperature was assumed to be constant.

**Table 4.** Parameters of the numerical analysis.

| Item | Condition |
|---|---|
| Time | Steady state |
| Viscous model | RSM turbulence model |
| Liquid | Water |
| Input quantity | Water 1 L<br>Alumina Powder 25 g |
| Particle density | 3950 Kg/m$^3$ |
| Particle size | 14.5 μm |
| Input flow rate | 0.35 m/s |
| Wall | No-slip condition (smooth walls) |
| Outlet | Overflow: pressure-outlet<br>Underflow: pressure-outlet |

## 3. Results and Discussion

### 3.1. Search for the Hydrocyclone Separator Design Factor Using FFD

The overflow rate percentage was set based on the results of the 12 runs generated by FFD, which were obtained by fabricating four hydrocyclone separators for comparison, as shown in Figure 7.

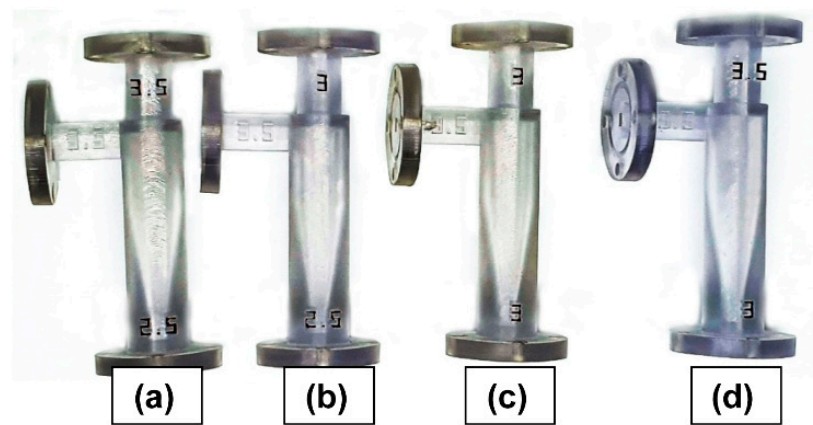

**Figure 7.** Four hydrocyclone separators fabricated for the FFD experiments: (**a**) OF: 3.5, UF: 2.5; (**b**) OF: 3, UF: 2.5; (**c**) OF: 3, UF: 3; (**d**) OF: 3.5, UF: 3 (OF: overflow; UF: underflow).

Experiments were conducted in response to the 12 generated runs. Table 5 summarizes the sampling conditions of FFD, the result values for the response variable, and the overflow rate percent.

**Table 5.** FFD run results.

| Standard Order | Run Order | Vortex Finder Diameter | Spigot Diameter | Overflow Rate Percent |
|---|---|---|---|---|
| 6 | 1 | 3.5 | 2.5 | 54 |
| 2 | 2 | 3.5 | 2.5 | 54 |
| 11 | 3 | 3.0 | 3.0 | 39 |
| 3 | 4 | 3.0 | 3.0 | 38 |
| 9 | 5 | 3.0 | 2.5 | 49 |
| 4 | 6 | 3.5 | 3.0 | 44 |
| 10 | 7 | 3.5 | 2.5 | 53 |
| 8 | 8 | 3.5 | 3.0 | 44 |
| 1 | 9 | 3.0 | 2.5 | 48 |
| 12 | 10 | 3.5 | 3.0 | 44 |
| 7 | 11 | 3.0 | 3.0 | 38 |
| 5 | 12 | 3.0 | 2.5 | 48 |

As shown in Figure 8, the values of the outlet flow rate ratio close to 1 were identified using a contour plot. The horizontal and vertical axes represent the diameters of the vortex finder and spigot, respectively, and their boundary area values are indicated by the dotted lines, where the overflow rate value in the FFD is 1. As the values found here have a proportional relationship with each other, they appear as a series of values rather than a single value.

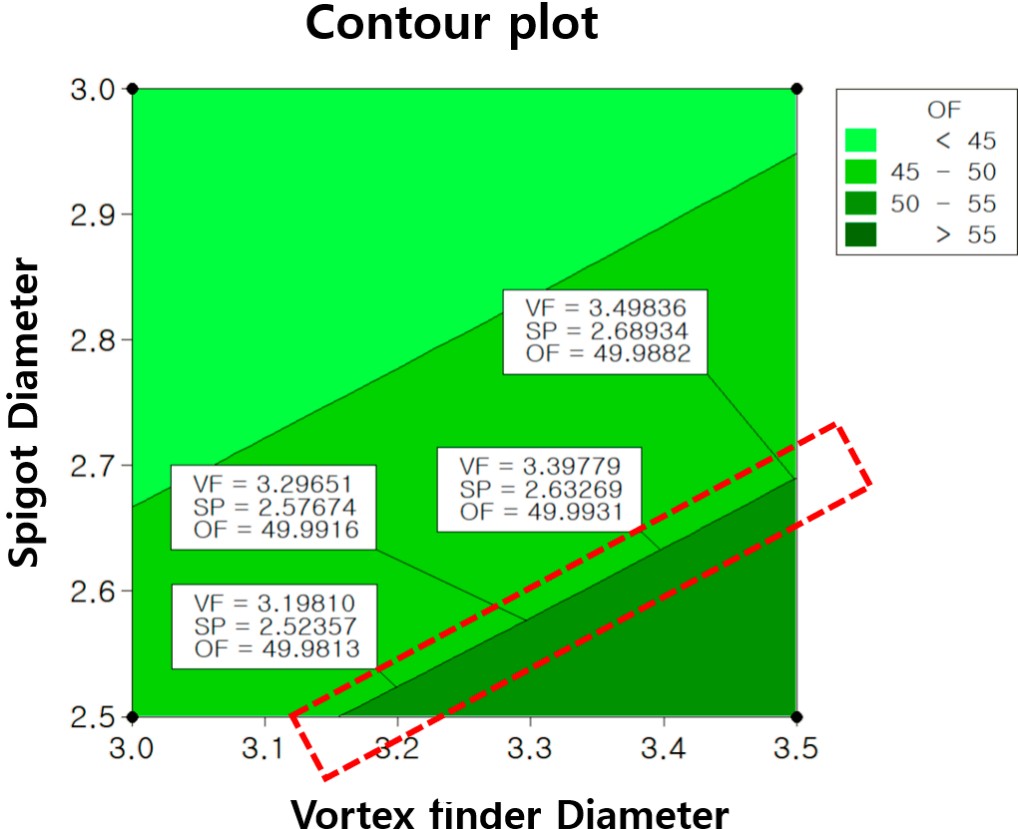

**Figure 8.** Contour plot for determining the outlet flow ratio, and the values that can determine the outlet flow ratio of 1 (indicated by the dotted lines).

Figure 9 shows the results of the contour plot as a 3D surface diagram, which indicates that the spigot diameter radically affects the overflow, whereas the vortex finder diameter has a gentle effect.

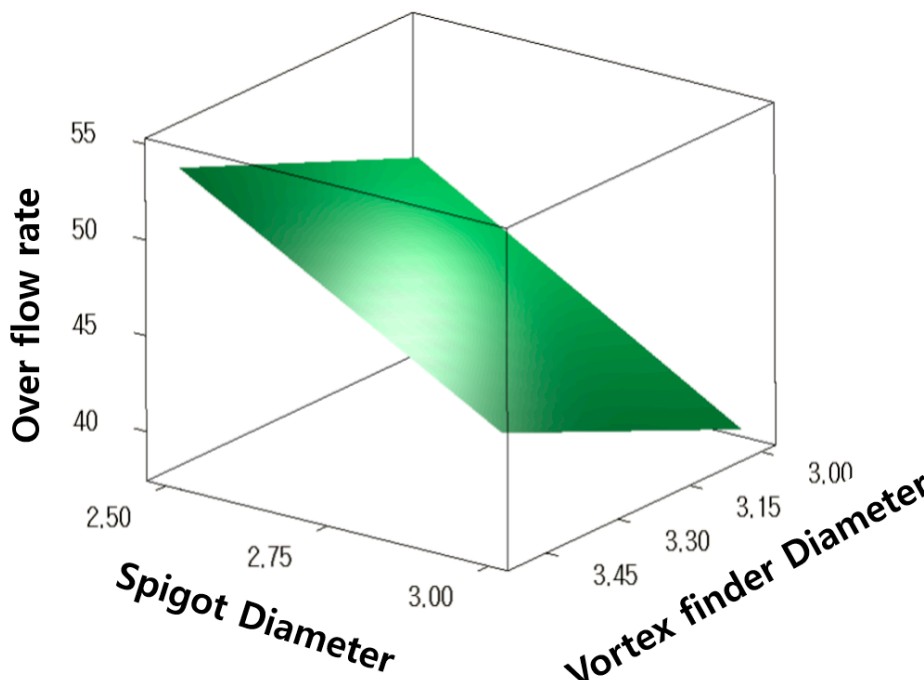

**Figure 9.** The spigot diameter affects more sensitively when checking the correlation of variables with the surface diagram.

The inlet dimensions were fixed, and the diameters of the two outlets were set as variables. The pressure change at the inlet was observed based on the changes in the outlet diameter when the flow rate was fixed. High pressure was observed in the inlet flow when the outlet diameter sizes were small, indicating better separation efficiency. Therefore, it is advantageous for the separation efficiency to select a size as small as possible if there are multiple points in the contour plot where the outlet flow ratio value of 1.

Considering this, the vortex finder and spigot diameters of 3.25 mm and 2.55 mm are considered suitable.

Furthermore, the optimization analysis results shown in Figure 10 reveal similar values, with vortex finder and spigot diameters of 3.250 mm and 2.5508 mm, respectively. The value in the second decimal place, where the judgment and analysis values overlapped, was selected as the optimal values, considering the 3D printing manufacturing precision and physical manufacturing error.

### 3.2. Numerical Results

Numerical analysis was performed using the Fluent software to investigate the adequacy of the vortex finder and spigot diameters optimized by FFD and confirm whether the FFD result was close to the actual outlet flow ratio value of 1.

Results showed that a cyclone was formed from the inlet through the vortex finder, due to which water collided with the inner wall of the cyclone body while descending toward the underflow outlet. As a result, dust collected at the lower spigot. Furthermore, a sufficient centrifugal force was obtained by increasing the swirl flow speed as the inner wall became narrower towards the lower section of the cylinder. In addition, the downward vortex passed through the distal end of the inner cylinder and grew larger as it approached the vertex of the spigot. Finally, the water was discharged as an overflow while rotating upward, owing to the occurrence of an ascending secondary vortex. The results are presented in Figure 11. The flow rates of the underflow and overflow were found to be approximately 0.00018 kg/s, with a higher underflow. The outlet flow ratio was 0.97. Although the value was 0.03 less than the target value of 1, the difference was not significant enough to affect the experimental results. Therefore, we concluded that the desired outflow ratio value

could be obtained by applying the values determined by FFD. Table 6 summarizes the numerical analysis result of the flow rate.

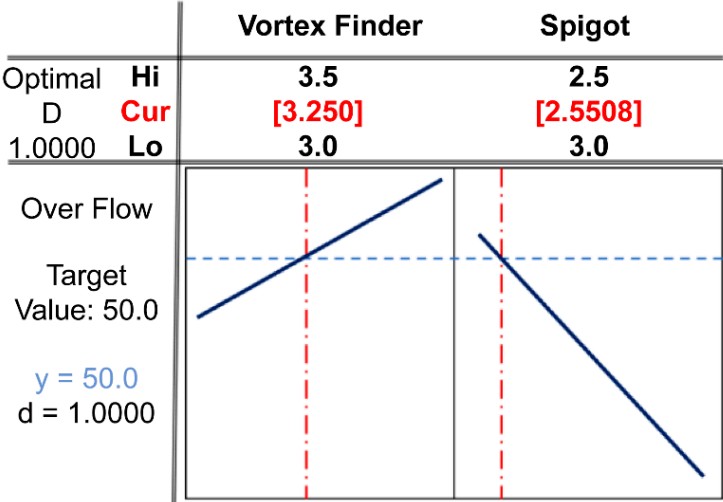

**Figure 10.** Optimization analysis result of the diameters of the vortex finder and spigot for optimizing the outlet flow ratio.

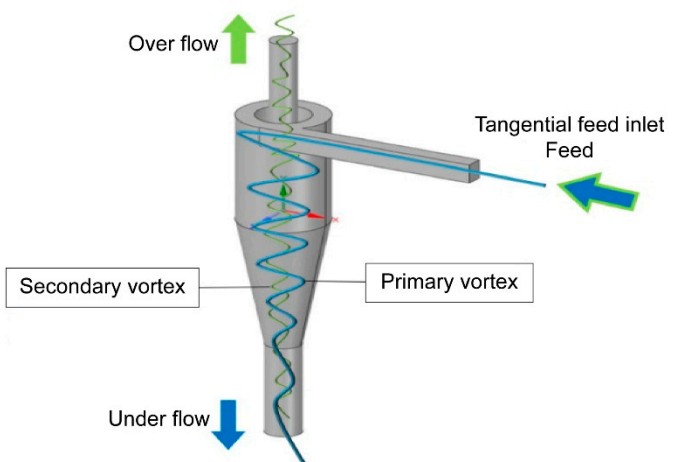

**Figure 11.** Flow direction of the optimized mini-hydrocyclone separator.

**Table 6.** Numerical analysis result of the flow rate of the outlets of the mini-hydrocyclone separator.

| Classification | Underflow | Overflow |
|---|---|---|
| Mass flow rate (Kg/s) | 0.00589 | 0.00571 |
| Flow rate | 51 | 49 |

Furthermore, numerical analysis was performed to determine whether the changes in the diameters of the vortex finder and spigot affected the separation flow of the particles. After adding 1690 $AL_2O_3$ particles, we obtained results for 1530 particles. The separation efficiency was found to be 94.77%, as 1450 particles emerged as underflow. As shown in Table 1, a separation efficiency of 93.67% was obtained for the experimental result before adjusting the outlet flow ratio. The 1% increase in separation efficiency can be attributed to the decrease in the vortex finder diameter [20]. As the outlet diameter decreases, the internal pressure of the cyclone increases. This results in an increase in internal pressure, which, in turn, leads to an increase in separation efficiency.

*3.3. Experiment Results*

The diameters of the vortex finder and spigot were optimized to 3.25 mm and 2.5 mm, respectively, by FFD, and the results were verified using numerical analysis. The acquired results were further confirmed by conducting experiments using the hydrocyclone separators with optimized values fabricated using 3D printing.

The purpose of this experiment was to identify whether we could achieve an outlet flow ratio of 1 from the two optimized outlet diameters and confirm the separation efficiency using the gravimetric method.

First, the flow rates from the two outlets were measured to be similar to that of the experimental results. Although the numerical analysis predicted the greater flow rate of the underflow, the amount of increase was insignificant. Therefore, it was not easily measurable in the actual observation, and almost the same flow rate was observed. Figure 12 shows the collection of fluids from the outlets in separate beakers.

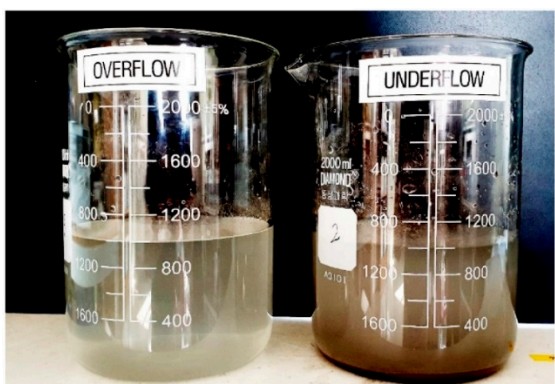

**Figure 12.** Sludge and water collected in separate beakers from the outlets after the separation efficiency measurement experiment.

Table 7 summarizes the experimental results. A targeted outlet flow ratio of 1 was achieved by obtaining the same flow rate from the two outlets. The efficiency was found to be 95.39% from the collected sludge using the gravimetric method.

**Table 7.** Experimental results using the optimized mini-hydrocyclone with an outlet flow ratio of 1.

| Total Overflow Volume | Total Underflow Volume | Overflow Sludge Weight | Underflow Sludge Weight | Separation Efficiency |
|---|---|---|---|---|
| 1000 mL | 1000 mL | 0.271 g | 46.961 g | 95.39% |

However, a difference of approximately 0.7% was observed compared with the result predicted by the numerical analysis, which is sufficiently reliable even when considering experimental errors. Furthermore, the separation efficiency increased by 1.72% compared with that of the originally designed mini-hydrocyclone separator. The experiment in this study was conducted by changing the outlet diameter while retaining the inlet dimensions, that is, when the flow rate was fixed, the changes in pressure at the inlet were observed based on the changes in the outlet diameter. A previous study reported that the separation efficiency of a cyclone separator improved as the inlet pressure increased. The sum of the two outlet diameters of the optimized mini-hydrocyclone separator was slightly decreased, compared with the reference model before the experiment, which can be attributed to the slight increase in the pressure in the inlet. Table 8 summarizes the three separation efficiencies measured using the gravimetric method.

**Table 8.** Separation efficiency results based on the changes in the outlet flow ratio determined by FFD.

| Classification | Outlet Flow Ratio | Separation Efficiency |
| --- | --- | --- |
| High-efficiency Stairmand model | 0.54 | 93.67% |
| Numerical analysis of the cyclone separator after adjusting the outlet flow ratio | 0.97 | 94.77% |
| Cyclone separator after adjusting the outlet flow ratio | 1.00 | 95.39% |

## 4. Conclusions

In this study, the separation efficiency of mini-hydrocyclone separators was measured using the gravimetric method. The separation efficiency of the originally designed mini-hydrocyclone separator with an outlet flow ratio value of 0.54 was calculated as 93.67%.

To apply the gravimetric method more objectively, the optimum values of the diameters of the vortex finder and spigot with an outlet flow ratio of 1 were determined using FFD. The desired outlet diameters could be selected with flexibility using a contour plot in the FFD. Moreover, the surface analysis results showed that the spigot diameter radically affected the overflow rate, while the vortex finder diameter had a more moderate effect.

The acquired results were verified through numerical analysis and experiments. The outlet flow ratio value changed from 0.54 to 1, whereas the separation efficiency increased from 93.67% to 95.39%.

An optimized mini-hydrocyclone separator was fabricated using the 3D printing method, and its performance was verified using the gravimetric method through experiments. The fabricated cyclone separators had an outlet flow ratio of 1 and exhibited a 1.72% increase in separation efficiency.

Based on the experimental and numerical analysis results of hydrocyclone, it was confirmed that the outlet flow ratio of the hydrocyclone can be predicted by the FFD method with an approximate error rate of 2%.

**Author Contributions:** Conceptualization, H.-W.Y. and M.-C.K.; methodology, H.-W.Y. and M.-C.K.; software, J.-Y.K. and Y.-W.L.; valida-tion H.-W.Y. and J.-Y.K.; formal analysis, H.-W.Y. and Y.-W.L.; investigation, H.-W.Y. and J.-Y.K.; resource, H.-W.Y. and J.-Y.K.; data curation, H.-W.Y.; writing—original draft preparation, H.-W.Y.; writing—review and editing, H.-W.Y.; visualization H.-W.Y.; supervision, M.-C.K.; project administration, H.-W.Y.; funding acquisition, H.-W.Y. All authors have read and agreed to the published version of the manuscript.

**Funding:** This research received no external funding.

**Institutional Review Board Statement:** Not applicable.

**Informed Consent Statement:** Not applicable.

**Data Availability Statement:** The datasets used and/or analyzed during the current study are available from the corresponding author on reasonable request.

**Acknowledgments:** The authors thank for the support from Hong Sung.

**Conflicts of Interest:** The authors declare no conflict of interest.

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
