# Peer review of "Optimization of the Outlet Flow Ratio of Mini-Hydrocyclone Separators Using the Full Factorial Design Method to Determine the Separation Efficiency"

_separations, doi:10.3390/separations8110210_

Round 1

Reviewer 1 Report

The paper entitled: „Optimization of the outlet-flow ratio of mini-hydro cyclone separators using the full factorial design method to determine the separation efficiency” deals with an experimental study in which the authors fabricated mini-hydro cyclone separators using the 3D printing method for application in the steam and water analysis system (SWAS) in a thermal power plant instead of the conventional strainer filter. I think, that the manuscript is an interesting paper however a lot improvements are necessary.

A list of comments follows:

  • An abstract need to state more clearly aim of your work and methodological approach as well.
  • It would need to identify what led to the design and selection of this segment of research and what was conducted in previous research that led to this.
  • Did the authors previously conduct any research that led to the design and selection of this research segment? This would need to be mentioned in the Introduction section.
  • It is also extremely important to emphasize the novel nature of the research.
  • Line 57: What specific research is it about?
  • Lines 100 – 123: This is not the right place for this type of information, it should be extended and moved to the Introduction section.
  • The properties of the test solution are missing.
  • Among other crucial problems, discussion is missing. Discussion should unambiguously express a comparison of the achieved results with the previous knowledge of the topic. It must make clear what is completely new in the presented results and where these results differ from the findings of other authors, and in what they coincide with the published opinions. Discussion should emphasise to the newly opened issues and the need for their solution. This is completely missing.

Author Response

[General review] Optimization of the outlet-flow ratio of mini-hydro cyclone separators using the full factorial design method to determine the separation efficiency” deals with an experimental study in which the authors fabricated mini-hydro cyclone separators using the 3D printing method for application in the steam and water analysis system (SWAS) in a thermal power plant instead of the conventional strainer filter. I think, that the manuscript is an interesting paper however a lot improvements are necessary.

 [Our response] We greatly appreciate the reviewer’s summary and evaluation of our manuscript.

Comment (1) An abstract need to state more clearly aim of your work and methodological approach as well. It would need to identify what led to the design and selection of this segment of research and what was conducted in previous research that led to this.

 Our response (1) Thank you for suggestion, accordingly, in the revised manuscript abstract has been modified.

[Modification of the manuscript]
"Abstract: Cyclone separators are widely used to eliminate particles flowing through pipelines in equipment from various industrial processes. Unlike general filters, cyclone separators can constantly and effectively eliminate foreign substances present in the fluid flowing through the equipment. In this study, we fabricated mini-hydro cyclone separators using the 3D printing method for application in the steam and water analysis system (SWAS) in a thermal power plant instead of the conventional strainer filter. The gravimetric method was used to measure the separation efficiency of the hydrocyclone separators and compare the weights of the sludge dis-charged from the underflow and overflow outlets. The outlet flow ratio was optimized by adjusting the diameters of the spigot and vortex finder of the separators, which influenced the out-let flow rate. To apply the gravimetric method more objectively, the optimum values of the diameters of the vortex finder and spigot with an outlet flow ratio of 1 were determined using full factorial design (FFD) in the design of the experiment (DOE). The obtained values were verified through numerical analysis using the ANSYS Fluent software. Furthermore, after fabrication of the mini hydro cyclone separators using an SLA-type 3D printer, we conducted a numerical analysis, and the results were compared to that of the actual experiment. It was observed that the use of FFD can effectively optimize the desired outlet flow ratio in the mini-hydrocyclone separator. In addition, the changes in the outlet flow ratio do not affect the separation efficiency of the cyclone separators. " on page 1, line 11-27

Comment (2) Did the authors previously conduct any research that led to the design and selection of this research segment? This would need to be mentioned in the Introduction section. It is also extremely important to emphasize the novel nature of the research.

Our response (2) Thank you for suggestion, accordingly, in the revised manuscript the introduction has been heavily modified. 

Comment (3) Line 57: What specific research is it about?

Our response (3) This refers to the first hydrocyclone that was designed. The position of the sentence has been moved to avoid confusion. 

[Modification of the manuscript]
 "Results revealed that the fabricated mini-hydrocyclone separators exhibited satisfactory separation efficiency of over 90%. However, the majority of water flow discharge came from the underflow outlet, where it was discarded." on page 4, line 132-134.

Comment (4) Lines 100 – 123: This is not the right place for this type of information, it should be extended and moved to the Introduction section.t is also extremely important to emphasize the novel nature of the research

Our response (4) Thank you for suggestion, accordingly, in the revised manuscript introduction has now been modified. 

[Modification of the manuscript]
 "Cyclone separators are mechanical devices that separate foreign substances from fluids using centrifugal force. They are widely used in household cleaners and air purifiers owing to their simple structure and can be manufactured easily as they do not require separate driving devices [1]. Cyclone separators can be classified as hydrocyclone and gas cyclone separators. Gas cyclone separators are used to separate solids and liquids from gases [2]. Hydrocyclone separators are used to separate liquids and solids. First, water comprising solid particles is injected through the inlet of the device, and the particles and water separate owing to the centrifugal force. The large and heavy particles are eliminated from the underflow outlet, whereas the small and light particles are eliminated from the overflow outlet [3].

Considering that the cooling tower and steam boiler in thermal power plants use water, the water quality is analyzed in real time using the steam and water analysis system (SWAS) to protect the equipment. Generally, the water samples received by SWAS from different equipment through pipelines are contaminated with foreign substances. Therefore, in this study, we develop a mini-hydrocyclone to replace the strainer-type filter used in the SWAS equipment in nuclear and thermal power plants to eliminate foreign substances flowing through the pipelines and prevent failure of the water quality analyzer to improve the overall analysis efficiency. Figure 1 is a schematic diagram of the SWAS of a thermal power plant. First, the process water send to the 1st chiller for cooling. Then, the water is supplied to the hydrocyclone to remove sludge. Finally, the water passes through a second chiller and is sent to the water quality analyzer.

In this study, we designed the mini-hydrocyclone separators based on the high-efficiency Stairmand model while focusing on changing other factors and retaining the diameter of the cylindrical body depending on the suggested flow rate. Figure 2 is a concept diagram of hydrocyclone. Several published studies have determined the performance of a cyclone separator from the relative proportions of its shape, based on indicators such as the inlet dimension, inlet pressure, spigot diameter, vortex diameter, cylinder diameter, and cone length [2,14]. " on page 1-2, line 31-58.

Comment (5) The properties of the test solution are missing. Among other crucial problems, discussion is missing. Discussion should unambiguously express a comparison of the achieved results with the previous knowledge of the topic. It must make clear what is completely new in the presented results and where these results differ from the findings of other authors, and in what they coincide with the published opinions. Discussion should emphasise to the newly opened issues and the need for their solution. This is completely missing.

Our response (5) Thank you for suggestion, accordingly, in the revised manuscript the conclusion has been modified as follows. 

[Modification of the manuscript]
 "In this study, we measured the separation efficiency of mini-hydrocyclone separators using the gravimetric method. The separation efficiency of the originally designed mini-hydrocyclone separator with an outlet flow ratio value of 0.54 was calculated as 93.67%.

To apply the gravimetric method more objectively, the optimum values of the diameters of the vortex finder and spigot with an outlet flow ratio of 1 were determined using FFD. The desired outlet diameters could be selected with flexibility using a contour plot in the FFD. Moreover, the surface analysis results showed that the spigot diameter radically affected the overflow rate, while the vortex finder diameter had a more moderate effect.

The acquired results were verified through numerical analysis and experiments. The outlet flow ratio value changed from 0.54 to 1, whereas the separation efficiency increased from 93.67% to 95.39%.

An optimized mini-hydrocyclone separator was fabricated using the 3D printing method, and its performance was verified using the gravimetric method through experiments. The fabricated cyclone separators had an outlet flow ratio of 1 and exhibited a 1.72% increase in separation efficiency.” on page 13, line 343-362.

Finally, we would like to extend our gratitude for your time and effort spent reviewing our paper.

Reviewer 2 Report

The paper needs a major revision, before it one can decide on an acceptance. The following points need attention:

1. In general, the matter is presented in a quite disordered and unstructured manner, which is quite difficult to follow. In general, a much better and more structured presentation is required. An example of this is that there are two sections with the same title and different numbers: „2.3. Numerical Analysis“ and „3.2 Numerical Analysis“ ?!

2.The results are not sufficiently and clearly discussed. An example is Figure 9. The figure is presented without any discussion. Furhermore, it is very difficult to understand, what is meant by the figure. The titles of the sub-figures are „Particle Inlet Velocity“, „Particle Top Velocity“ and „Particle Bottom Velocity“. What do they mean, at all ? Furthermore, there are positive and negative values in the scale. What is the direction, what kind of velocity is this ?

3. The description of the mathematical/numerical flow is not satisfactory. What was the grid resolution ? Was a grid independence study performed ? What kind of discretization schemes used ?

Was the turbulent particle dispersion considered, by which method, etc.

4. Furthermore the statement „The flow inside the hydrocyclone was mixed with laminar and turbulent flows, and both the water and particles were considered as a continuum flow. Therefore, a Reynolds stress model (RSM) was used for the analysis.“ sounds strange for two reasons.

Firstly, what does RSM have to do with the mentioned features ?

Secondly, if individual particles trajectories are calculated, one cannot say that the particle flow is considered to be a continuum flow.

5. The introduction shall be supported by an additional conceptual figure of a cyclone and the figure shall be utilized in explaining the phenomenon, which would help to better understanding.

6. In explainin things, the „we“ form is used too frequently („we did this“, „we did that“ etc.). This is not the generally preferred scientific style. Exceptionally, it can be used sometimes, in certain situations. However, in general, the „passive“ form shall be used, instead („this was done“, „that was done“, etc.).

7. There are many senetences that don’t sound well thought. For example, let us take the sentence on line 43: „Considering cooling and heat exchange in thermal power plants …“.

Is „cooling“ not a „heat exchange“ ?

8. There are many style errors and inconsistencies in the style. Examples: „overflow“ and „underflow“ are sometimes written as „over flow“ and „under flow“. „Sludge“ is written sometimes with a capital initial and sometimes as „sludge“ with lower case initial. In the section headings, the initials are sometimes capitalized, sometimes not. Line 115: A sentence should not start with a number (50 g …).

9. The language usage is not good. The help of a native speaker or professional help is recommended.

Author Response

[General review] The paper needs a major revision, before it one can decide on an acceptance. The following points need attention.

[Our response] We greatly appreciate the reviewer’s summary and evaluation of our manuscript.

Comment (1) In general, the matter is presented in a quite disordered and unstructured manner, which is quite difficult to follow. In general, a much better and more structured presentation is required. An example of this is that there are two sections with the same title and different numbers: „2.3. Numerical Analysis“ and „3.2 Numerical Analysis“ ?!

Our response (1) Thank you for this suggestion, accordingly, in the revised manuscript the suggested title and document structure has been modified as follows.

[Modification of the manuscript]
"2.1 Mini Hydrocyclone Fabrication and Experiment” on page 4."2.3 Numerical Methods” on page 7."3.2 Numerical Results” on page 10."3.3 Experiment Results” on page 11.  

Comment (2) The results are not sufficiently and clearly discussed. An example is Figure 9. The figure is presented without any discussion. Furhermore, it is very difficult to understand, what is meant by the figure. The titles of the sub-figures are „Particle Inlet Velocity“, „Particle Top Velocity“ and „Particle Bottom Velocity“. What do they mean, at all ? Furthermore, there are positive and negative values in the scale. What is the direction, what kind of velocity is this ?

Our response (2) In Figure 9. Has been deleted in the revised version because the exact meaning could not be clearly expressed owing to the picture resolution problem.What we wanted to show in the picture was the flow direction of the particles.

The flow in the top direction of the hydrocyclone indicates a positive value, and the value flowing to the bottom indicates a negative value.

Comment (3) The description of the mathematical/numerical flow is not satisfactory. What was the grid resolution ? Was a grid independence study performed ? What kind of discretization schemes used ?

Our response (3) The number of grids in the numerical study was 111,708.For the numerical analysis of complex flow, transient state and the Reynolds stress model (RSM) for turbulence were used.VOF was selected for multiphase flow interpreting models.As for numerical analysis conditions, semi-implicit pressure linked equations (SIMPLE) were used for the pressure-velocity coupling scheme and the pressure staggering option (PRESTO) was used for the pressure. Second-order upwind scheme was used for solver setting.In order to predict the separation efficiency of the hydrocyclone and the movement of particles, discrete phase models (DPM) were used.

[Modification of the manuscript]
 "A numerical analysis was performed using the ANSYS Fluent software to verify the acquired results and match the outlet flow ratio of 1 in the FFD. The flow inside the hydro-cyclone was mixed with laminar and turbulent flows. For the numerical analysis of complex flow, transient state and the Reynolds stress model (RSM) for turbulence were used. Among many available turbulence models, RSM is an appropriate model for interpretation of flow with strong swirl.

Internal flow of the hydrocyclone represented a condition where water, particles and air coexisted. Therefore, VOF was selected among multiphase flow interpreting models. And both the water and particles were considered as a continuum flow. Tracking is realized by the writing force equilibrium equation of a solid particle assuming different types of force according to complexity of the model. This approach is only suitable when a reasonable number of particles is tracked and the suspension is diluted (max 5-10% mass fraction of the solid phase)." on page 7, line 201-213.

 Comment (4) Furthermore the statement „The flow inside the hydrocyclone was mixed with laminar and turbulent flows, and both the water and particles were considered as a continuum flow. Therefore, a Reynolds stress model (RSM) was used for the analysis.“ sounds strange for two reasons. Firstly, what does RSM have to do with the mentioned features ? Secondly, if individual particles trajectories are calculated, one cannot say that the particle flow is considered to be a continuum flow.

Our response (4) Thank you for suggestion, the manuscript has been extensively revised in accordance with your recommendations.. 

[Modification of the manuscript]
" A numerical analysis was performed using the ANSYS Fluent software to verify the acquired results and match the outlet flow ratio of 1 in the FFD. The flow inside the hydro-cyclone was mixed with laminar and turbulent flows. For the numerical analysis of complex flow, transient state and the Reynolds stress model (RSM) for turbulence were used. Among many available turbulence models, RSM is an appropriate model for interpretation of flow with strong swirl.

Internal flow of the hydrocyclone represented a condition where water, particles and air coexisted. Therefore, VOF was selected among multiphase flow interpreting models. And both the water and particles were considered as a continuum flow. Tracking is realized by the writing force equilibrium equation of a solid particle assuming different types of force according to complexity of the model. This approach is only suitable when a reasonable number of particles is tracked and when the suspension is diluted (max 5-10% mass fraction of the solid phase)." on page 7, line 201-213.

Comment (5) The introduction shall be supported by an additional conceptual figure of a cyclone and the figure shall be utilized in explaining the phenomenon, which would help to better understanding.

 Our response (5) Accordingly, Figures 1 and 2 have been included in the revised version to address your point.

[Modification of the manuscript]

Figure 1. SWAS schematic diagram with cyclone separator applied.  :on page 2, line 79.

 Figure 2. Hydrocyclone design concept and parameters.  : on page 3, line 98.

Comment (6) In explainin things, the „we“ form is used too frequently („we did this“, “we did that“ etc.). This is not the generally preferred scientific style. Exceptionally, it can be used sometimes, in certain situations. However, in general, the „passive“ form shall be used, instead („this was done“, „that was done“, etc.).

 Our response (6) Thank you for suggestion, the manuscript has been revised in accordance with your recommendations. 

[Modification of the manuscript]
 "In this study, the separation efficiency of mini hydrocyclone separators was measured using the gravimetric method. The separation efficiency of the originally designed mini-hydrocyclone separator with an outlet flow ratio value of 0.54 was calculated as 93.67%.

To apply the gravimetric method more objectively, the optimum values of the diameters of the vortex finder and spigot with an outlet flow ratio of 1 were determined using FFD. The desired outlet diameters could be selected with flexibility using a contour plot in the FFD. Moreover, the surface analysis results showed that the spigot diameter radically affected the overflow rate, while the vortex finder diameter had a more moderate effect.

The acquired results were verified through numerical analysis and experiments. The outlet flow ratio value changed from 0.54 to 1, whereas the separation efficiency increased from 93.67% to 95.39%.

An optimized mini-hydrocyclone separator was fabricated using the 3D printing method, and its performance was verified using the gravimetric method through experiments. The fabricated cyclone separators had an outlet flow ratio of 1 and exhibited a 1.72% increase in separation efficiency

Based on the experimental and numerical analysis results of hydrocyclone, it was confirmed that the outlet flow ratio of the hydrocyclone can be predicted by FFD method with an approximate error rate of 2%." on page 13, line 343-363.

Comment (7) There are many senetences that don’t sound well thought. For example, let us take the sentence on line 43: „Considering cooling and heat exchange in thermal power plants …“.

Is „cooling“ not a „heat exchange“ ?

 Our response (7) Yes, we agree with your comments and would like to thank you for your critical review of the manuscript. We changed the ‘cooling’ to a ‘cooling tower’. And the ‘heat exchange’ has changed to ‘Steam Boiler’. We included the gas turbine combine cycle schematic diagram below.  

[Modification of the manuscript]
“ Considering that the cooling tower and steam boiler in thermal power plants use water, the water quality is analyzed in real time using the steam and water analysis system (SWAS) to protect the equipment.” on page 1, line 41-43.

Comment (8) There are many style errors and inconsistencies in the style.

Examples: “overflow“ and “underflow“ are sometimes written as “over flow“ and “under flow“. “Sludge“ is written sometimes with a capital initial and sometimes as “sludge“ with lower case initial. In the section headings, the initials are sometimes capitalized, sometimes not. Line 115: A sentence should not start with a number (50 g …).

Our response (8) ) Thank you for your keen observations, these mistakes and flaws have now been corrected in the revised version as follows.

[Modification of the manuscript]
" The mixture was mixed using 50 g  with a particle size distribution of 14.5 μm in 2000 ml of water using an agitator. " on page 4, line 112.
"Total overflow volume” on page 5, Table 1.

"Total underflow volume” on page 5, Table 1.

"Overflow sludge weight” on page 5, Table 1.

"Underflow sludge weight” on page 5, Table 1.

"Pareto chart of the standardized effects on the overflow of (A) Vortex finder and (B) Spigot diameter confirmed by Pareto chart." on page 6, Figure 6.

"Overflow” on page 6, Table 2.

"Underflow” on page 6, Table 2.

"Overflow” on page 7, Table 4.

"Underflow” on page 7, Table 4.

"Four hydrocyclone separators fabricated for the FFD experiments. (a) OF: 3.5, UF: 2.5, (b) OF: 3, UF: 2.5, (c) OF: 3, UF: 3, and (d) OF: 3.5, UF: 3 (OF: Overflow; UF: Underflow). " on page 8, Figure 7.

"Overflow” on page 8, Table 5.

"Underflow” on page 11, Table 6.

"Overflow” on page 11, Table 6.

"Total overflow volume” on page 12, Table 7.

"Total underflow volume” on page 12, Table 7.

"Overflow sludge weight” on page 12, Table 7.

"Underflow sludge weight” on page 12, Table 7.

Comment (9) The language usage is not good. The help of a native speaker or professional help is recommended.

Our response (9) Thank you for your feedback, we would like to confirm that we have had the manuscript checked to improve language. We attached the certificate of English editing as follows. 

Finally, we would like to extend our gratitude for your time and effort spent reviewing our paper.

Round 2

Reviewer 1 Report

The authors have corrected the critical remarks suggested in the review. 
The paper is now in better condition than before - the corrections make difference. I recommend the paper for publication in the Separations journal.

Author Response

We would like to extend our gratitude for your time and effort spent reviewing our paper.

Reviewer 2 Report

I think that the authors are not very familiar with mathematical modelling. The VOF model is not convenient for interpenetrarting continua. On the contrary, the VOF model is convenient for separated two-phase flows, where there is a clear interface between the phases, which is of interest.

Furthermore, it is not clear to me, how air enters into the cyclone. Does input to hydrocylcone contain also air ? This is actually a point which cannot be stated incidentally, but needs a clear elaboration, demonstration and discussion. Water/air fields with interfaces etc. would need to be shown and discussed, at least.

Author Response

Reviewer #2:

[General review] I think that the authors are not very familiar with mathematical modelling.

The VOF model is not convenient for interpenetrarting continua.

On the contrary, the VOF model is convenient for separated two-phase flows, where there is a clear interface between the phases, which is of interest.

[Our response] We greatly appreciate the reviewer’s summary and evaluation of our manuscript.

Comment (1) Furthermore, it is not clear to me, how air enters into the cyclone. Does input to hydrocylcone contain also air ? This is actually a point which cannot be stated incidentally, but needs a clear elaboration, demonstration and discussion.  Water/air fields with interfaces etc. would need to be shown and discussed, at least.

Our response (1) Thank you for this suggestion, accordingly, in the revised manuscript has been modified as follows.  

[Modification of the manuscript]
" Air core formation has been investigated in many hydrocyclones. Hydrocyclones form a central air core which extends over the complete hydrocyclone length. Air is sucked in the core at the underflow discharge. In addition, Formation of the air core in any hydrocyclone is considered to be an indication of vortex stability[20]. " on page 7, line 206-209.

[Our additional response]

Thank you for your feedback.To improve the language, we will once again have the manuscript corrected by a native speaker.And, We will attach again the certificate of English editing  

Finally, we would like to extend our gratitude for your time and effort spent reviewing our paper.
